# Macrophage-P2X4 receptors pathway is essential to persistent inflammatory muscle hyperalgesia onset, and is prevented by physical exercise

**Graciana de Azambuja**[1], **Fernando Moreira Simabuco**[2], **Maria Cláudia Gonçalves de Oliveira**[1]*

1 Universidade Estadual de Campinas (UNICAMP), Faculdade de Ciências Aplicadas, Laboratório de Estudos em Dor e Inflamação (LABEDI), Limeira, São Paulo, Brasil, 2 Universidade Estadual de Campinas (UNICAMP), Faculdade de Ciências Aplicadas, Multidisciplinary Laboratory in Food and Health, Limeira, São Paulo, Brasil

* mfusaro@unicamp.br

## Abstract

Peripheral inflammation may lead to severe inflammatory painful conditions. Macrophages are critical for inflammation; modulating related pathways could be an essential therapeutic strategy for chronic pain diseases. Here we hypothesized that 1) Macrophage-P2X4 receptors are involved in the transition from acute to persistent inflammatory muscle hyperalgesia and that 2) P2X4 activation triggers a pro-inflammatory phenotype leading to Interleukin-1β (IL-1β) increase. Once physical exercise prevents exacerbated inflammatory processes related to chronic diseases including chronic muscle pain, we also hypothesized that 3) physical exercise, through PPARγ receptors, prevents P2X4 receptors activation. With pharmacological behaviour, biomolecular analysis and swimming physical exercise in a mouse model of persistent inflammatory muscle hyperalgesia we demonstrated that P2X4 receptors are essential for transitioning from acute to persistent inflammatory muscle hyperalgesia; Phosphorylation of p38MAPK indicated P2X4 signalling activation associated with inflammatory macrophage and an increase of IL-1β expression in skeletal muscle; Exercise-PPARγ receptors prevented phosphorylation of p38MAPK in muscle tissue. Our findings suggest that exercise-PPARγ modulates the acute inflammatory phase of developing persistent muscle hyperalgesia by controlling p38MAPK-related P2X4 signalling. These highlight the great potential of modulating macrophage phenotypes and P2X4 receptors to prevent pain conditions and the ability of physical exercise to prevent inflammatory processes related to chronic muscle pain.

## 1. Introduction

In injury/noxious stimulated tissues macrophages are activated to a pro-inflammatory phenotype. Seeking balance and tissue repair, anti-inflammatory macrophages are induced to resolve the inflammatory state [1–3]. There are several mechanisms underlying these macrophage inflammatory and anti-inflammatory activation. Of particular interest, the

**Data availability statement:** All relevant data are within the manuscript and its Supporting information files.

**Funding:** São Paulo Research Foundation (FAPESP) – process number: 2018/13599-1; 2020/10585-0; 2021/02921-2 Coordination of Superior Level Staff Improvement (CAPES) – 001. The funders had no role in study design, data collection and analysis, decision to publish, or preparation of the manuscript.

**Competing interests:** The authors declare no competing interests.

Purinergic 2X purinoceptor 4 (P2X4) are ligand-gated cation channel, activated in response to ATP binding, commonly expressed on the membrane of both central [4,5] and peripheral immune cells [6,7]. The increase in the extracellular concentration of ATP following a noxious stimulus activates P2X4 receptors, triggers calcium influx [8] and the phosphorylation of p38 MAPK (Mitogen-Activated Protein Kinase) [9]. The P2X4 receptors have been related to hyperalgesia and allodynia in several pain models [10–13]. However, it remains unknown whether macrophage-P2X4 in skeletal muscle is enrolled in the transition acute-to-persistent inflammatory muscle pain and, if so, whether it could represent a pathway to be modulated, providing immunological beneficial effects, preventing and or treating persistent inflammatory muscle pain.

Peripheral mechanisms of persistent inflammatory muscle pain onset are still understudied, in addition to understanding prevention/treatment effects related to physical exercise. Physical exercise effects on chronic pain conditions have been increasingly associated with good benefits to the pathology. Longitudinal epidemiological studies indicate that physically active people are less likely to develop chronic pain conditions throughout their lives [14,15] which stimulates studies to uncover new mechanisms of exercise effects on pain pathology. Among the most studied, opioids and serotoninergic are key mechanisms that reduce hyperalgesia in non-inflammatory pain models [16,17]. Furthermore, we and others demonstrated that physical exercise reduces and/or prevents pain conditions in different animal models [17–21]. However, few studies tackle the effects of physical exercise on inflammatory pathways related to the development of skeletal muscle pain specifically.

Regular physical exercise is important for general health and well-being in humans. One of the main effects occurs through the adaptation of the skeletal muscle to the physical effort, where the exercise practice influences several cell types in the tissue, responding to metabolic demands and modulating other organs systemically [22]. Thus, the effects of exercise on inflammatory and metabolic pathways in skeletal muscle have been investigated in different types of chronic diseases [22–24], evidencing the benefits of physical exercise-related mechanisms in muscle tissue. In this sense, in our recent studies, we showed that regular physical exercise prevents acute [21] and inflammatory-related chronic muscle hyperalgesia [25] by mechanisms dependent on neuroimmune modulation. We found that 1) macrophages are involved in the transition from acute to chronic inflammatory muscle hyperalgesia; 2) inflammatory ("M1-like") macrophages are increased during the acute phase, as well as the IL-1β release in muscle tissue; 3) regular physical exercise previous to the inflammatory insult trigger anti-inflammatory effects, by preventing the persistent muscle hyperalgesia and the increase in IL-1β through activation of the peroxisome proliferator-activated receptor gamma (PPARγ), in addition to decreasing "M1-like" and inducing anti-inflammatory ("M2-like") macrophages in the muscle tissue [25]. Taken together with previous studies [19,26,27], these results pointed out the macrophages as key cells involved in the transition to chronic muscle hyperalgesia. The question that remains open is related to initial events that trigger such inflammation-related macrophage effects. Also, we speculate that physical exercise-induced PPARγ activation is a potential preventive pathway that could be enrolled in macrophages' pro-/anti-inflammatory response.

In the present study, we hypothesized that P2X4 receptors expressed on muscle-macrophages, contribute to initial inflammation that is related to the transition from acute to persistent inflammatory muscle pain, in a model of persistent inflammatory hyperalgesia. Therefore, the prevention of persistent muscle hyperalgesia by physical exercise could involve modulation of the P2X4 signaling pathway through PPARγ receptors. We tested these hypotheses using behavioral sensory pharmacology, western blotting, Enzyme Linked Immuno Sorbent Assay (ELISA) and immunofluorescence.

## 2. Materials and methods

### 2.1. Animal care

For in vivo experiments, we used male Swiss mice (*Mus Musculus*) weighing 25–35 grams at the beginning of the experiments. Animals were provided by the institutional Multidisciplinary Research Center of the Institute of Biology (CEMIB/UNICAMP). The procedures were approved by the local institutional ethics committee for the use of animals (CEUA-UNICAMP, protocol number: 5244-1/2018) and were performed according to the National Council for the Control of Animal Experimentation (CONCEA, Brazil) and the research committee guidelines and ethics of the International Association for the Study of Pain in Conscious Animals [28]. All animals were housed in enriched plastic cages (up to five animals per cage) containing wood shavings and plastic tubes, following a 12-hour light/dark cycle (light on at 7:00 am). Food and water were *ad libitum*, except during experimental procedures. The experimental sessions were carried out during the light phase, from 9:00 am to 5:00 pm, in a quiet room with a controlled temperature ($\pm23$ °C). The animals were randomly assigned to experimental groups (Fig 1A) and the researchers were blinded to the experimental groups in all behavioral experiments.

### 2.2. The model of acute and persistent mechanical inflammatory muscle hyperalgesia

In this study, we investigated an experimental model of persistent mechanical inflammatory muscle hyperalgesia induced by two injections, standardized in gastrocnemius of Swiss mice [29]. To induce acute muscle hyperalgesia, a transient inflammatory process was triggered through the intramuscular injection of $\lambda$ – carrageenan (Cg, 100 µg/muscle) into the belly of the gastrocnemius muscle. After 10 days, when animals demonstrated that the nociceptive threshold was at baseline levels, the second stimulus (PGE$_2$; 1 µg/muscle) was injected through the same site to trigger longer and more intense muscular hyperalgesia, referred as persistent muscle hyperalgesia.

To assess hyperalgesia, a mechanical stimulus was applied to the injected muscle. The hyperalgesic threshold was assessed at different times: 1) baseline (before muscle injection); 2) on the day of the injection (day 0) after 1, 3 and 6 hours; 3) daily until the 10th day after the injection; 4) immediately before the PGE$_2$ injection (day 10, as a second baseline); 5) on the day of the injection (day 10) after 1 and 4 hours; 6) on days 11, 12 and again on day 17 (Fig 1B) [25].

To investigate the mechanism underlying the induction of acute and consequent persistent hyperalgesia, intramuscular pharmacological strategies were administered prior to carrageenan. To investigate whether the mechanisms were supporting persistent inflammatory muscle hyperalgesia, pharmacological strategies were administered prior to PGE$_2$. As a control to hyperalgesia and sensitization effects from the carrageenan, the control groups received isotonic saline (0.9% NaCl) instead of carrageenan, followed by PGE$_2$.

### 2.3. Mechanical testing of muscle nociceptive threshold to assess the muscle hyperalgesia

To assess muscle hyperalgesia, we used the Randall-Selitto analgesimeter (Insight, Brazil), which evaluates the change in nociceptive threshold through mechanical pressure. Before initial tests, animals were adapted to the restraint method and the apparatus. The equipment applies a gradual increasing linear pressure on the gastrocnemius muscle until leg withdrawal or vocalization occurs, allowing researchers to measure nociceptive thresholds.

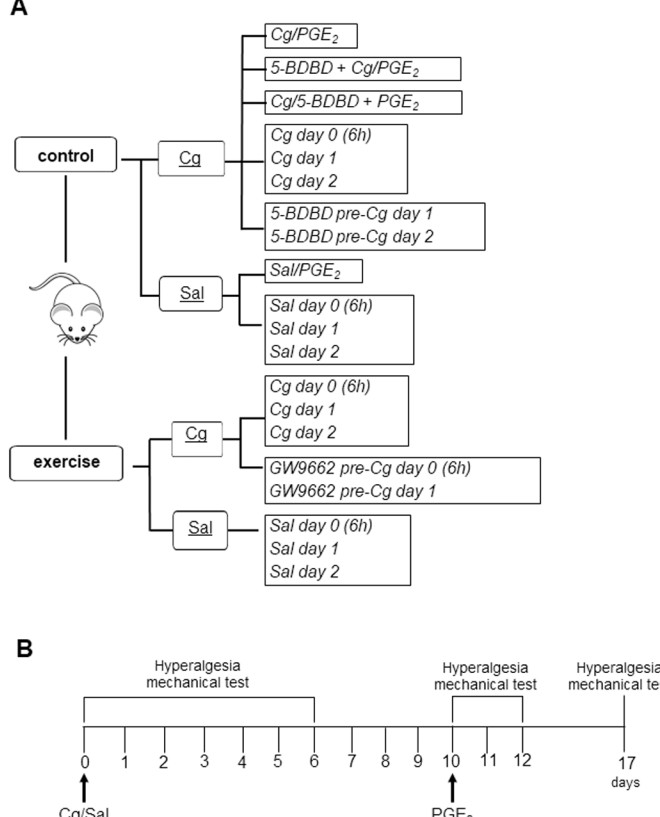

**Fig 1. Flowchart of procedures and experimental groups organization.** A) The animals were randomly divided into control and exercise groups (in bold), then into two subgroups with intramuscular injections of carrageenan (Cg) or saline (Sal) (underlined). Thereafter, subgroups underwent intramuscular injection of prostaglandin $E_2$ (PGE$_2$) (Sal/PGE$_2$, Cg/PGE$_2$, Exercise Sal and Exercise Cg) or additional pharmacological strategies that were previously administered to Cg/Sal or PGE$_2$. Additional groups received Cg or Sal injections and were euthanized at day 0 (6h), day 1 and day 2 (in italic). Final experimental groups are represented in black-lined boxes in italic. B) Experimental procedures related to the time course of the hyperalgesic mechanical tests performed before and after injections of Cg or Salt, and PGE$_2$. The mechanical test was Applied at baseline and from day 0 (1, 3 and 6h) until day 6 after injection of Cg or Sal. PGE$_2$ was injected on the 10th day, in the same muscle, and the mechanical test was applied on days 10 (1 and 4h), 11, 12 and 17. The pharmacological antagonist strategy (in bold) was performed before the first injections or the second injection.

Cutoff pressure was set at 50 g. The rounded tip with 2 mm in diameter evokes the nociceptive threshold of deep tissues [30]. Using the average of three measurements performed at 5-minute intervals, the nociceptive threshold was established [25,31]. The measurements taken at each time of interest after carrageenan/saline and PGE$_2$ injections were subtracted from the baseline measurements to define levels of mechanical inflammatory muscle hyperalgesia, being presented as delta (in grams) on the y-axis by increasing values [25,31].

## 2.4. Procedures for intramuscular injections, drugs, and doses

The right gastrocnemius muscle was used for intramuscular injections that were performed with a 50 μL Hamilton syringe connected to a 30-gauge needle. The final volume of injections was 20 μL/muscle. The following drugs were used: λ-carrageenan (100 μg/muscle; [29]), prostaglandin $E_2$ (PGE$_2$, 1 μg/muscle, [29]), the antagonist of PPARγ, GW9662

(2-chloro-5-nitro-N-phenylbenzamide, 9 ng/muscle, [25]) and the selective P2X4 receptor antagonist, 5-BDBD (5-(3-Bromophenyl)-1,3-dihydro-2H-Benzofuro[3,2-e]-1,4-diazepin-2-one, 50 μM/muscle [7]). Lambda Carrageenan was freshly prepared at a concentration of 1 mg/mL and subsequently diluted to a working solution of 5 μg/μL in 0.9% isotonic saline. The stock solutions of GW9662 and 5-BDBD were dissolved in dimethyl sulfoxide (DMSO), and of PGE$_2$ in 10% ethanol, according to the manufacturer's instructions, then resuspended in saline to reduce the DMSO concentration (maximum 0.05%) and ethanol (maximum 1%). All drugs were purchased from Sigma-Aldrich (St. Louis, MO, USA) and dissolved in isotonic saline (0.9%) to the working concentration.

## 2.5. Regular physical exercise protocol

Regular physical exercise was performed through a previously tested swimming protocol with a water temperature maintained at 31 ± 1 °C [25]. Before starting the exercise period, animals underwent a water adaptation protocol ([25], Table 1).

Regular physical exercise was performed for 15 days divided into three periods of 5 consecutive days with a two-day break between each period. The intensity was controlled by a progressive increase in session volume and a decrease in passive pauses ([25], Table 2). When the animal's swimming behavior was shown to be altered to inappropriate movements, such as floating, "climbing", diving, and swimming without direction, the animal was removed from the water.

## 2.6. Procedure with primary culture of peritoneal macrophages

Intraperitoneal macrophages were used to investigate the involvement of P2X4 receptors in the induction of an inflammatory phenotype. The animals were anesthetized with 5% isoflurane and then euthanized with an overdose. Then, 5 mL of PBS-EDTA (5 mM) was injected into the peritoneal cavity and the abdomen was massaged for one minute. The PBS-EDTA solution containing cells was collected from the peritoneal cavity with the same syringe and transferred to a sterile tube, which was centrifuged (300 × g/3 minutes). The supernatant was removed, and the remaining cells were washed with PBS again by resuspension in PBS (5 mL) followed by centrifugation. The cells were resuspended in 1 mL of cRPMI (RPMI-1640 with 0.3 g/L L-glutamine, 100 U/mL of penicillin and streptomycin, amino acid [100×], 1 U/mL of sodium pyruvate and 10% fetal bovine serum [FBS]; Merck, Darmstadt, Germany). Cells were plated in a 12-well plate (1 × 10$^6$ cells/ well) and incubated for 24 hours (37 °C, 5% CO$_2$). The non-adherent cells were washed with PBS.

Macrophages were treated and analyzed by immunofluorescence to assess whether treatment with carrageenan and/or 5-BDBD induced polarization in cells. For this, cells were treated with carrageenan (Cg, 100 μg/mL) alone, or with 5-BDBD (2 μM) for 6 hours. Control cells were incubated at the same time with either cRPMI and PBS alone or 5-BDBD alone. The wells were washed with PBS and the cells were fixed with 4% paraformaldehyde (4% PFA), blocked with 3% bovine serum albumin (3% BSA) and 3% donkey serum, and incubated

**Table 1. Adaptation protocol to the liquid environment.**

| Days | Water level | Time |
|---|---|---|
| 1 | 1 cm | 5 min |
| 2 | 1 cm | 10 min |
| 3 | 30 cm | 5 min |
| 4 | 30 cm | 10 min |
| 5 | 30 cm | 15 min |
| 6 | 30 cm | 20 min |

**Table 2. Physical exercise protocol of regular swimming in Swiss mice.**

|  | Days | Session | Pauses (min) | Series | Training volume (min) | Total (min) |
|---|---|---|---|---|---|---|
| Phase 1 | 1 | 5 × 10 | 10 | 5 | 50 | 90 |
|  | 2 | 5 × 10 | 10 | 5 | 50 | 90 |
|  | 3 | 3 × 10/ 1 × 20 | 10 | 4 | 50 | 80 |
|  | 4 | 2 × 15/ 2 × 10 | 10 | 4 | 50 | 80 |
|  | 5 | 2 × 15/ 2 × 10 | 10 | 4 | 50 | 80 |
| Phase 2 | 6 | 1 × 20/ 1 × 20/ 1 × 10 | 7 | 3 | 50 | 70 |
|  | 7 | 1 × 20/ 1 × 20/ 1 × 10 | 7 | 3 | 50 | 70 |
|  | 8 | 1 × 25/ 1 × 25 | 7 | 2 | 50 | 60 |
|  | 9 | 1 × 25/ 1 × 25 | 7 | 2 | 50 | 60 |
|  | 10 | 1 × 25/ 1 × 25 | 7 | 2 | 50 | 60 |
| Phase 3 | 11 | 1 × 30/ 1 × 20 | 4 | 2 | 50 | 60 |
|  | 12 | 1 × 30/ 1 × 20 | 4 | 2 | 50 | 60 |
|  | 13 | 1 × 40/ 1 × 10 | 4 | 2 | 50 | 60 |
|  | 14 | 1 × 40/ 1 × 10 | 4 | 2 | 50 | 60 |
|  | 15 | 1 × 40/ 1 × 10 | 4 | 2 | 50 | 60 |
| Phase 4 | 16 | 1 × 50 | 0 | 1 | 50 | 50 |
|  | 17 | 1 × 50 | 0 | 1 | 50 | 50 |
|  | 18 | 1 × 50 | 0 | 1 | 50 | 50 |
|  | 19 | 1 × 50 | 0 | 1 | 50 | 50 |
|  | 20 | 1 × 50 | 0 | 1 | 50 | 50 |

overnight with primary antibodies: rat anti-mouse F4/80 (clone Cl:A3-1) (1:1000; BioRad, MCA497G) Armenian hamster Monoclonal CD11c antibody (1:100, Abcam, ab33483) and CD206 Mouse Monoclonal Mannose Receptor antibody (1:50; Abcam, ab8918). Secondary antibodies were donkey Alexa Fluor 488 anti-rat (1:250, Jackson ImmunoResearch), goat Alexa Fluor 555 anti-Armenian Hamster (1:250, Jackson ImmunoResearch) and goat Dylight 405 anti-mouse (1:250, Jackson ImmunoResearch). Negative slides were incubated with secondary antibodies only, allowing detection of nonspecific fluorescence. Images were obtained using a fluorescence microscope (Leica, MDI6000b) with a × 40 objective lens and processed with ImageJ (NIH). Co-labeled cells were manually counted by more than one researcher, all blinded to the experimental groups.

## 2.7. Procedures with RAW 264.7 macrophage culture

To investigate the involvement of P2X4 receptors in the inflammatory activation of macrophages, we tested the receptor signaling pathway by the p38 MAPK phosphorylation. Murine macrophage strain RAW 264.7 was cultured in Dulbecco's Modified Eagle's Medium (DMEM), supplemented with 10% heat-inactivated FBS and 100 U/mL of penicillin and streptomycin (Gibco). To verify the relationship between macrophage polarization and P2X4 gene expression, RAW 264.7 cells, in a 12-well plate, were stimulated with Lipopolysaccharides (LPS, 1 μg/mL for 4 hours). To verify if an anti-inflammatory stimulus would prevent the macrophage polarization and the P2X4 gene expression, RAW 264.7 cells were incubated with interleukin-4 (IL-4) for 24 hours, before LPS stimulation. In all experiments involving lineage, the sample n. represented in the analysis groups are composed of, at least, three different passages of the cells.

Total RNA was isolated from cell culture using TRI reagent® (Sigma-Aldrich) following the manufacturer's instructions. RNA quality and concentration were quantified by NanoDrop (Thermo Scientific). A total of 1 μg of RNA was considered for treatment with

Amplification Grade DNAse I (Thermo Scientific). 500 ng of DNAse-treated RNA was used for the preparation of complementary DNA (cDNA) strands with the High-capacity cDNA Reverse Transcription Kit (Applied Biosystems). Real-time quantitative polymerase chain reaction (PCR) was performed with the Master Mix for PCR, SYBR® Green (Applied Biosystems), and the relative gene expression was calculated using the delta-delta Ct method, normalized to gene expression of TATA-binding protein (*Tbp*) [32]. The assay was performed on a Quant Studio 6 real-time PCR system (Applied Biosystems). The gene expression represented in the graphs is treatment samples compared to control samples (fold-change over control, y-axis). The primer sequences are described in Table 3.

For protein analysis of p38 MAPK and p-p38 MAPK, cells were plated in 6-well plates and, after 24 hours, stimulated with carrageenan (100 μg/mL) for 30 minutes (previously standardized time, S3 Fig). To investigate the involvement of P2X4 receptors on p-p38 MAPK expression, the 5-BDBD antagonist was added to the culture 30 minutes before carrageenan. Cells were lysed and protein extracted using a lysis buffer (2% SDS, 10% glycerol, 125 mM Tris-HCl pH 6.8, 1× protease and phosphatase inhibitor). Lysates were carefully transferred to a tube and homogenized by sonicator. Total protein was quantified using the Pierce BCA protein Assay kit (Thermo Fisher) following the manufacturer's instructions, with an initial dilution of 1:5.

Protein lysates were separated by SDS-polyacrylamide gel electrophoresis (12%) and transferred for 1 hour to a nitrocellulose membrane in an ice-cold buffer of 20% methanol. Membranes were blocked with 5% skim milk, or 5% bovine albumin (BSA) for phosphorylated proteins, in Tris-buffered saline and 0.1% Tween-20® and incubated overnight at 4 °C with antibodies against p38 MAPK (1:1000, Cell Signaling, #9212), p-p38 MAPK$^{Thr180/Tyr182}$ (1:1000, Cell Signaling, #9211), and beta tubulin (1:1000, Cell Signaling, #2146). After washing, membranes were incubated at room temperature for 1 hour with their respective secondary antibodies conjugated to horseradish peroxidase (HRP). Blots were visualized by enhanced chemiluminescence on a ChemiDoc Imaging System (Bio-Rad). Original unaltered membranes are shown in S4 and S5 Figs.

## 2.8. Analysis of muscle tissue protein expression by western blotting

On day 0 (6h after injections), day 1 and day 2 after injections, mice were euthanized, the injected gastrocnemius muscle (right) was extracted, placed in microtubes and immersed in liquid nitrogen until storage at −80°C. Muscle weight was adjusted to be similar across all samples. The samples were homogenized with the beadruptor and sonicator, with 1mL of ice-cold lysis buffer with protease and phosphatase inhibitor (Roche, Switzerland, catalog number: 11697498001; 4906845001) [29]. Total protein was quantified using the Pierce BCA protein Assay kit (Thermo Fisher) following the manufacturer's instructions, in initial dilutions of 1:20.

On SDS-polyacrylamide gel (12%), protein lysates were separated by electrophoresis and transferred to a nitrocellulose membrane in an ice-cold 20% methanol buffer for 1 hour.

**Table 3. Primers' sequences used for quantitative PCR analysis.**

| Protein | Gene | Forward sequence | Reverse sequence |
|---|---|---|---|
| CD86 | *Cd86* | AAC TTA CGG AAG CAC CCA CG | CGT CTC CAC GGA AAC AGC AT |
| Argenin 1 | *Arg1* | CTT GCG AGA CGT AGA CCC TG | TCC ATC ACC TTG CCA ATC CC |
| IL-1β | *Il1b* | GAA ATG CCA CCT TTT GAC AGT G | TGG ATG CTC TCA TCA GGA CAG |
| P2X4 | *P2rx4* | TCA TCC GCA GCC GTA AAG TG | ACA CGA ACA CCC ACC CAA TG |
| TNF-α | *Tnf* | CAG GCG GTG CCT ATG TCT C | CGA TCA CCC CGA AGT TCA GTA G |
| TATA *Binding protein* | *Tbp* | ACC CTT CAC CAA TGA CTC CTA TG | TGA CTG CAG CAA ATC GCT TGG |

Blocking was done with 5% skim milk or 5% bovine albumin (BSA) for phosphorylated proteins, in Tris-buffered saline and 0.1% Tween-20® and incubated overnight at 4 °C with the same antibodies listed above, against p38 MAPK (1:1000, Cell Signaling) and p-p38 MAPK (1:1000, Cell Signaling). After washing, membranes were incubated at room temperature for 1 hour with their respective secondary antibodies conjugated to horseradish peroxidase (HRP) and blots were visualized by enhanced chemiluminescence on a ChemiDoc Imaging System (Bio-Rad). All membranes were stained with Ponceau S dye for the total protein evaluation and original images are presented in S1 Fig.

### 2.9. Analysis of muscle tissue cytokine expression

The right gastrocnemius muscle was extracted, placed in microtubes and snap-frozen in liquid nitrogen until storage at − 80 ºC. For homogenization, each sample weight was adjusted to be similar and 750 μL of ice-cold lysis buffer (phosphate-buffered saline (PBS), 0.05% Tween 20® and protease inhibitor cocktail (Roche, Switzerland, catalog number: 11697498001) was added into a safe-lock microtube. Mechanical lysis was performed with two rounds of bead ruptor (40 s, high intensity) followed by two rounds with an ultrasonic homogenizer (10% amplitude, 10 s). To separate the supernatant, the homogenate was centrifuged at $10,000 \times g$ for 15 min at 4 ºC [33]. Total protein concentration was measured from the supernatant by Bradford assay, with an optimal dilution of 1:20 (Sigma-Aldrich, catalog number: B6916). The cytokine interleukin 1 beta (IL-1β) was quantified by ELISA kit (catalog numbers: 431414, Biolegend) according to the manufacturer's instructions, with a dilution of 1:5 and duplicated samples for all groups in the same plate to ensure comparison under the same test conditions. The final cytokine concentration is presented as pg/mg of total protein.

### 2.10. Statistical analysis

Kolmogorov-Smirnov test indicated that data followed a normal Gaussian distribution, thus parametric tests were applied. To verify significant differences between the groups' means, one-way ANOVA, two-way ANOVA or Student's $t$ test (unpaired or paired, when applicable) were used. After variance analysis with one-way and two-way ANOVA, post-hoc contrasts were performed using Tukey's test. For the behavior time course, an area under the curve (AUC ± standard error (SE)) analysis was performed to assess the effects of interventions on acute and persistent hyperalgesia. Acute period was considered from day 0 to day 3 after carrageenan injection (AUC Δ,g × 3 days) and persistent period AUC was considered from day 10 to 17 (AUC Δ,g × 10–17 days).

The size of the sample group for continuous variables was determined by estimating the population standard deviation and the magnitude of the difference between the means of the groups [34], and they are described in the legends of the figures. The samples are composed of categorical variables (sedentary, exercised, drug treatment groups in general, genotype, etc.). Continuous quantitative data values are expressed as mean ± standard error of the mean (SEM). Outliers were calculated using GraphPad Quick calc tool which performs the Grubbs' test. Significance level was set to 0.05. All data were analyzed by GraphPad Prism 9.0 software. For all tests, the significance was set at $p < 0.05$.

## 3. Results

### 3.1. P2X4 receptors in skeletal muscle are involved in the transition of acute to persistent inflammatory muscle hyperalgesia

In a previous study, we demonstrated that macrophages are essential cells to induce persistent inflammatory muscle hyperalgesia [25]. Since P2X4 receptors in the skeletal muscle are

expressed in macrophages [7], here we aimed to investigate the involvement of P2X4 receptors in the transition of acute to persistent inflammatory muscle hyperalgesia. To this end, we injected the selective P2X4 receptor antagonist, 5-BDBD, into the muscle to locally block the receptor in the acute or persistent phase.

Acute and persistent inflammatory muscle hyperalgesia were partially reversed when 5-BDBD (50 µM/muscle) was injected previously to carrageenan in the same muscle (5-BDBD+Cg/PGE$_2$) when compared to the carrageenan control group (Cg/PGE$_2$) (Fig 2, $p <0.01$). However, there is still a significant difference when compared to the saline control group (Sal/PGE$_2$) (Fig 2A, $p < 0.05$). The injection of 5-BDBD (50 µM/muscle) prior to PGE$_2$ in a muscle previously sensitized by carrageenan (Cg/5-BDBD+PGE$_2$) did not demonstrate significant reductions in the persistent muscle hyperalgesia when compared to the carrageenan control group (Fig 2A, $p <0.05$). The analysis of AUC demonstrated similar results (Fig 2B and 2C, acute: $p <0.0001$; persistent: $p <0.0001$). These results demonstrated that, in skeletal muscle tissue, P2X4 receptors are involved in the transition of acute to persistent inflammatory muscle hyperalgesia. Therefore, we were deep in the investigation of mechanisms in a cell-specific manner.

### 3.2. Inflammatory stimuli induce macrophage polarization via activation of P2X4 receptors

In our previous study, we demonstrated that activation of macrophage inflammatory phenotype seems to be a key factor in inducing persistent inflammatory muscle hyperalgesia since these activated cells were preferentially expressed in the acute phase of the muscle inflammation (from day 1 to 3 after carrageenan injection) [25]. In addition, we also observed at day 2 (but not day 1) an increased IL-1β expression in the muscle tissue after carrageenan injection

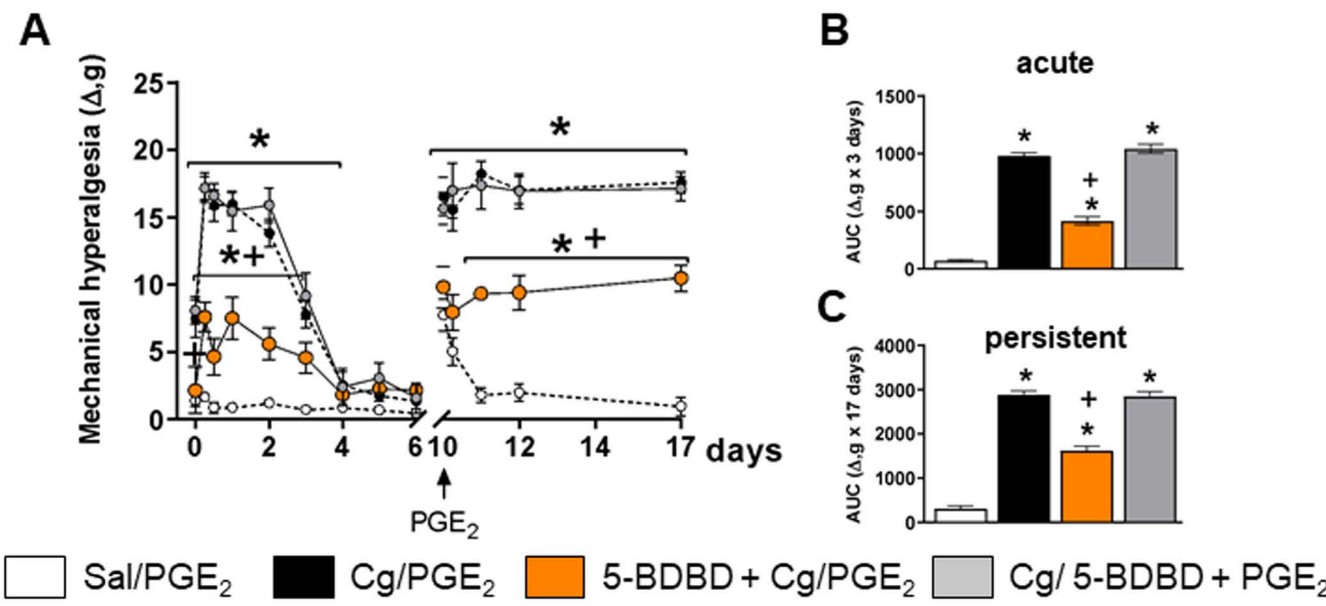

**Fig 2. P2X4 receptors are involved on the induction of acute and persistent muscular hyperalgesia.** A) Timeline graph showing acute muscle hyperalgesia after carrageenan (Cg) followed by persistent muscle hyperalgesia after PGE2 in the groups that received saline (Control Sal), carrageenan (Control Cg), and 5-BDBD antagonist before carrageenan (5-BDBD+Cg/PGE$_2$) or PGE$_2$ (Cg/5-BDBD+PGE$_2$) (two-way ANOVA, Tukey post hoc; F (39, 234) = 7.220). B and C) Area under the curve (AUC) of acute (B) and chronic (C) phases of Control Sal, Control Cg, 5-BDBD+Cg/PGE$_2$ and Cg/5-BDBD+PGE$_2$ (one-way ANOVA, Tukey's post hoc; F (3, 20) = 54.35, $p <0.0001$). For all graphs, the symbol "*" indicates difference for the Control Sal group and "+" indicates difference with the Control Cg group. N = 5 for 5-BDBD groups; N = 9 for the Control Cg group; N = 5 Control Group Sal.

[25], a hallmark of inflammatory macrophages activation [3,7,35]. Since P2X4 in skeletal muscle is involved in the induction of persistent inflammatory muscle hyperalgesia, we aimed to investigate whether activation of P2X4 receptors could be involved in macrophage polarization to the inflammatory phenotype.

Challenging primary peritoneal macrophages with carrageenan induced a greater proportion of F4/80[+] cells co-localized with CD11c[+] (F4/80[+]/CD11c[+], M1-like phenotype) than with PBS or 5-BDBD alone (Fig 3A and 3B, p <0.01). This proportion was decreased in macrophages challenged with 5-BDBD plus carrageenan (5-BDBD + Cg) but was still significantly higher than in PBS control cells (Fig 3B, p = 0.0265). Challenging primary macrophages with carrageenan did not increase the proportion of F4/80[+] cells co-localized with CD206[+] (F4/80[+]/

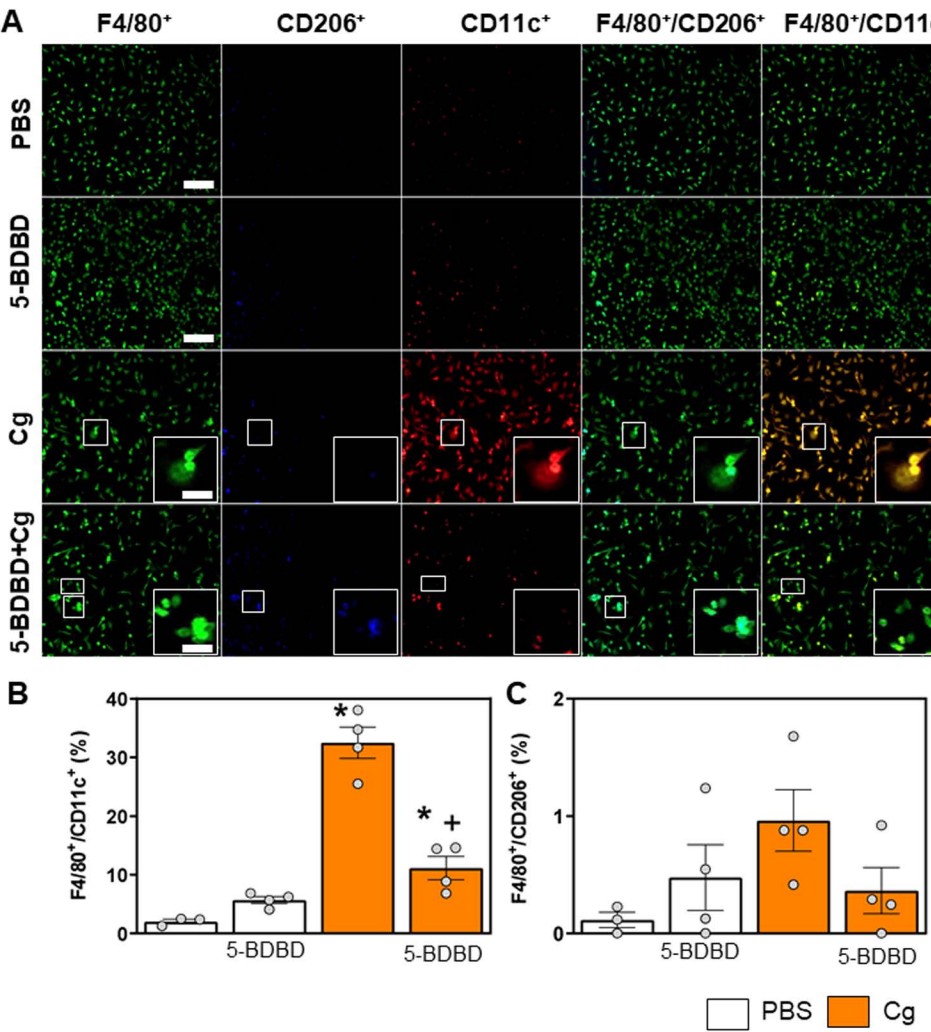

**Fig 3. Carrageenan-stimulated peritoneal macrophages acquire an inflammatory phenotype, that is prevented by P2X4 blockade.** A) Representative images of the F4/80[+] marker indicating macrophage cells (green), CD11c[+] marker (red) and CD206[+] marker (blue) from PBS and 5-BDBD control cells, and cells stimulated with carrageenan (Cg) and with 5-BDBD and Cg (5-BDBD + Cg). Scale bar = 50 μm. White squares indicate the inset location. B and C) Analysis of the proportion of CD11c (B) and CD206 (C) labeling in relation to F4/80 labeling. B) Carrageenan stimulation increases the expression of CD11c[+] in F4/80[+] cells and pretreatment with 5-BDBD decreases it (one-way ANOVA, Tukey post hoc (B) $F_{(3, 11)}$ = 56.24; (C) $F_{(3, 11)}$ = 2.242). For all graphs, the symbol "*" indicates difference for the PBS group and "+" indicates difference for the Cg group. N = 4 to 5 for each group.

CD206[+], M2-like phenotype, Fig 3A and 3C, p = 0.2206). These results indicate that carrageenan induces an increase in macrophage activation towards an inflammatory phenotype through pathways involving P2X4 receptors.

In the same sense, challenging RAW 264.7 macrophages with the pro-inflammatory stimuli, LPS, modulated inflammatory-related gene expression. There was an increase in *cd86* (M1 macrophage marker), *arg 1* (M2 macrophage marker), *il1b* (IL-1β), and *p2rx4* (P2X4) genes when compared to PBS (Fig 4A, p <0.0001). The challenge with LPS plus IL-4, an anti-inflammatory stimulus [36], prevented the increase in inflammatory-related genes *cd86, il1b, tnf* and *p2rx4,* and induced an increase in anti-inflammatory-related gene *arg1* when compared to LPS control group (Fig 4A, p <0.0001). These results confirmed that

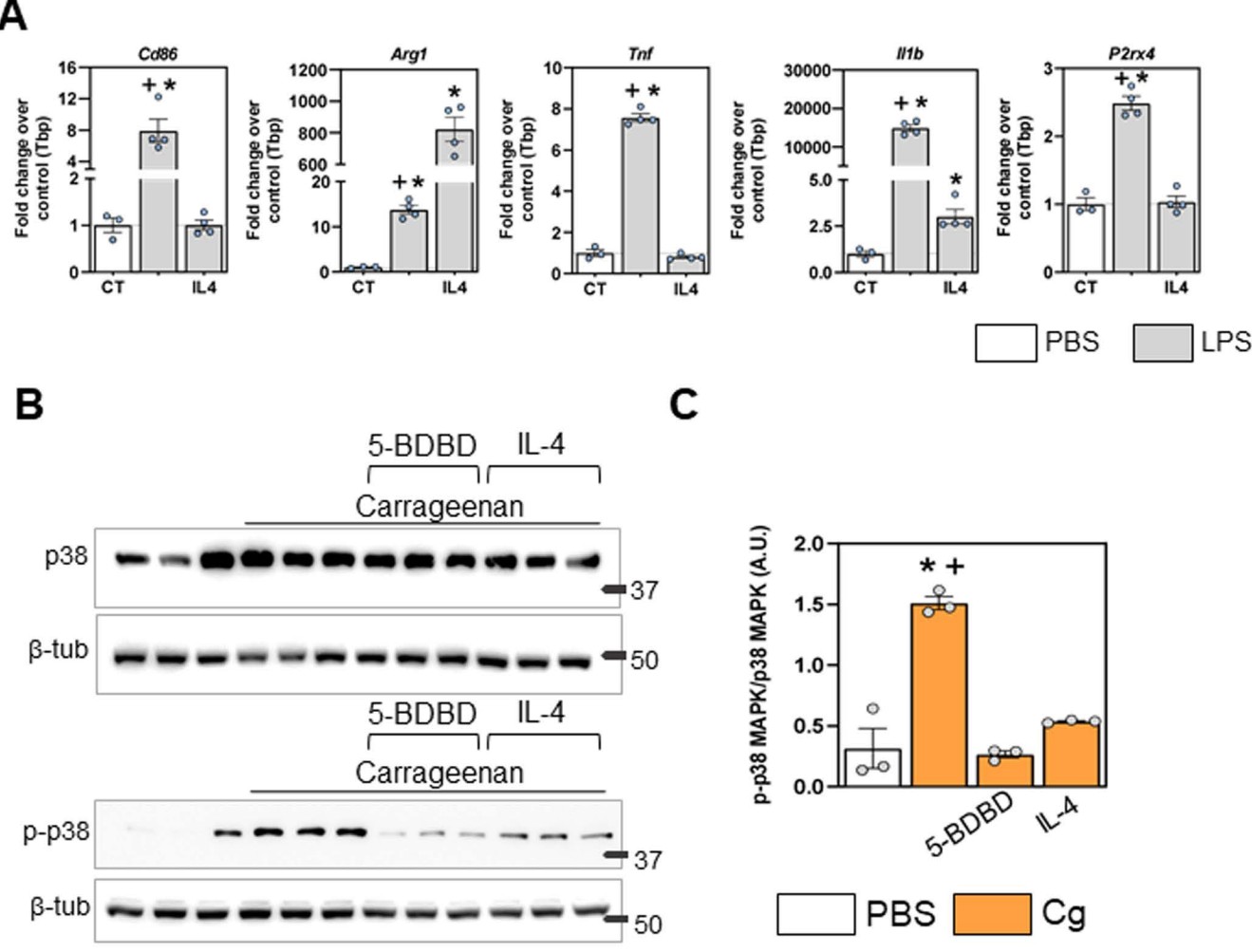

**Fig 4. P2X4 receptors are enrolled in macrophage inflammatory pathway activation.** A) Gene expression of cd86, arg1, tnf, il1b and p2rx4 in RAW 264.7 macrophages stimulated with LPS, LPS + IL-4 or PBS (one-way ANOVA, Tukey post hoc; $F_{(2, 8)} = 18.80$; $F_{(2, 8)} = 97.12$; $F_{(2, 8)} = 251.5$; $F_{(2.8)} = 84.27$, respectively). "*" indicates difference for the PBS group, "+" indicates difference with the IL-4 group. B) Western blotting demonstrating the expression of p38 MAPK, p-p38 MAPK and endogenous beta tubulin proteins, in RAW 264.7 treated with PBS, carrageenan (Cg), 5-BDBD + Cg or IL-4 + Cg. The number marking indicates the height of the marker in kilodaltons (kDa). C) Analysis of p-p38 MAPK/p38 MAPK expression (arbitrary units [A.U.]). Cg-treated cells express more p-p38 MAPK compared to other treatments. Blockage of P2X4 by 5-BDBD or anti-inflammatory stimuli IL-4, before Cg stimulation decreases p-p38 MAPK expression (one-way ANOVA, Tukey post hoc, $F_{(3, 8)} = 33.93$. For all graphs, the symbol "*" indicates difference for the PBS group, "+" indicates difference with IL-4 + Cg or 5-BDBD + Cg groups. N = 3 to 4 for each group.

pro-inflammatory and anti-inflammatory stimulation modulates the macrophage phenotype and that pro-inflammatory stimuli increase the expression of inflammation-related genes, along with increased *p2rx4* gene expression.

Next, we aimed to investigate the downstream protein signaling of P2X4 receptor activation in these cells. To this end, we challenged RAW 264.7 cells with carrageenan, 5-BDBD plus carrageenan or IL-4 plus carrageenan and evaluated the expression of p38 MAPK protein phosphorylation (p-p38 MAPK). The expression of p-p38 MAPK in RAW 264.7 was increased in cells challenged with carrageenan when compared to control PBS cells (Fig 4B and 4C, p < 0.0001). When P2X4 receptors were blocked by the 5-BDBD before carrageenan challenging, the p-p38 MAPK expression was decreased (Fig 4B and 4C, p < 0.0001). In addition, when macrophages were challenged with IL-4 + Cg, the increase in p-p38 MAPK was also prevented (Fig 4B and 4C, p < 0.0001). These results demonstrated that inflammatory stimulus induces activation of p38 MAPK through phosphorylation (p-p38MAPK expression) in macrophages, involving P2X4 activation.

Considering the *in vitro* results, we aimed to investigate whether the p-p38 MAPK pathway would be activated in our *in vivo* model, and whether P2X4 receptors are involved in this process. We evaluated the acute period timepoints (day 0 at 6 hours, day 1 and 2 after carrageenan injection) since p38 MAPK phosphorylation is an acute event [11] and, in this model, we previously observed an increase in inflammation ("M1-like" macrophages) from day 1, and the IL-1β increase at day 2 after injection of carrageenan [25].

Results in mice demonstrated that, in carrageenan control group (Cg), p-p38 MAPK was increased at day 0 (6 hours, Fig 5A and 5B, p < 0.0001) and day 1 (Fig 5C and 5D, p < 0.0001)

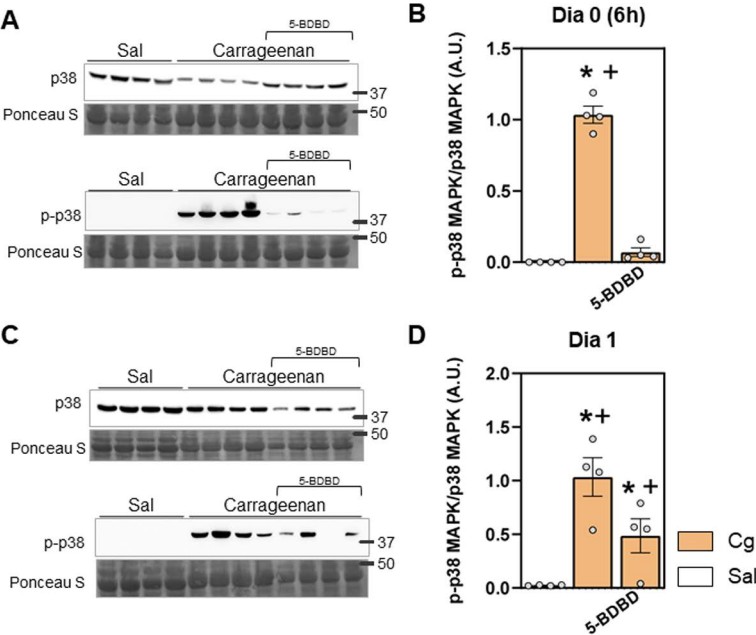

**Fig 5. P-p38 MAPK expression induced by Carrageenan in skeletal muscle involves P2X4 receptors.** A and C) Representative image of western blotting membranes demonstrating expression of p38 MAPK, p-p38 MAPK proteins and total proteins (Ponceau S) staining on day 0 (6 hours) (A) and day 1 (C), in saline (Sal), carrageenan (Cg) and Cg with the 5-BDBD strategy. The number marking indicates the height of the marker in kilodaltons (kDa). B and D) Quantification of the expression of p-p38 MAPK/p38 MAPK (arbitrary units [A.U.]) on day 0 (6 hours) (B) and day 1 (D) (two-way ANOVA, Tukey post hoc). "*" indicates significant difference (p < 0.05) with Sal and "+" significant difference (p < 0.05) with Cg group. Numbers are presented as Mean ± Standard Error (SE). N = 4 for each group.

after carrageenan injection. Interestingly, administration of P2X4 antagonist, 5-BDBD (50 μM/muscle), prior to carrageenan prevented p-p38 MAPK day 0 and day 1 (Fig 5C and 5D [day 0], and Fig 6C and 6D [day 1]; p < 0.0001, for all), suggesting the involvement of P2X4 receptors in this signaling process induced by carrageenan inflammation. Therefore, day 0 and day 1 after carrageenan injection were considered target inflammatory timepoints with P2X4 activation and to evaluate physical exercise effects. At day 2, p-p38 MAPK expression is not prevented by the blockage of P2X4 receptors before the carrageenan (S2A–2E Fig, p = 0.2037), however it is important to highlight that a protective effect of the P2X4 blockage is observed since IL-1β levels are reduced at day 2 after carrageenan and P2X4 antagonist administration (S2F Fig).

Taken together the *in vitro* and *in vivo* results, we demonstrate that carrageenan-induced inflammation activates an inflammatory state of macrophages, where P2X4 receptors are enrolled through p38 MAPK phosphorylation signaling, as evidenced by the pharmacological strategy. The 5-BDBD treatment indicates that P2X4 receptors are involved in the inflammation-induced intracellular signaling through phosphorylation of p38 MAPK *in vitro* and in muscle tissue. Considering this, we analyzed whether physical exercise's protective and anti-inflammatory effects on the skeletal muscle could be related to this signalling pathway.

### 3.3.  Physical exercise prevents p38 MAPK phosphorylation in acute inflammatory muscle hyperalgesia via PPARγ receptors

In our previous study, we demonstrated that regular physical exercise prevents the transition from acute to persistent inflammatory muscle hyperalgesia and modulates macrophage phenotype and cytokines through PPARγ activation in the acute phase [25]. Now, we aimed to investigate whether physical exercise would be able to modulate the p38 MAPK phosphorylation related to the inflammatory process through PPARγ receptors. In exercised animals (Ex) that received carrageenan, the increase in p-p38 MAPK was prevented at day 0 and day 1 (Fig 6A and 6B [day 0], and Fig 6C and 6D [day 1]; p < 0.0001, for all). The administration of the PPARγ receptor antagonist, GW9662 (GW, 9 ng/muscle), prior to carrageenan inhibited this prevention (Fig 6A and 6B [day 0], and Fig 6C and 6D [day 1]; p < 0.0001, for all). These results indicate that physical exercise modulated p38 MAPK phosphorylation during the inflammatory process through PPARγ receptors activation in muscle tissue.

### 4.  Discussion

Our findings provide significant insights into the mechanisms underlying the transition from acute to persistent inflammatory muscle hyperalgesia, emphasizing the central role of P2X4 receptors in macrophage-mediated inflammatory processes and the therapeutic potential of physical exercise in modulating these pathways.

We demonstrated that P2X4 receptors play a pivotal role in the progression of acute to persistent inflammatory muscle hyperalgesia. Pharmacological inhibition of P2X4 with the selective antagonist 5-BDBD resulted in a partial reversal of hyperalgesia when administered prior to carrageenan injection. This finding highlights the importance of P2X4 activation in the initial stages of the inflammatory response. However, the incomplete reversal of hyperalgesia indicates the involvement of additional mechanisms or pathways.

We demonstrated the transition from acute to persistent inflammatory muscle hyperalgesia involves the activation of P2X4 receptors in pro-inflammatory macrophages and a signaling pathway through p38 MAPK phosphorylation and an increase in IL-1β gene expression. When exploring the link between P2X4 activation and macrophage polarization, in vitro experiments

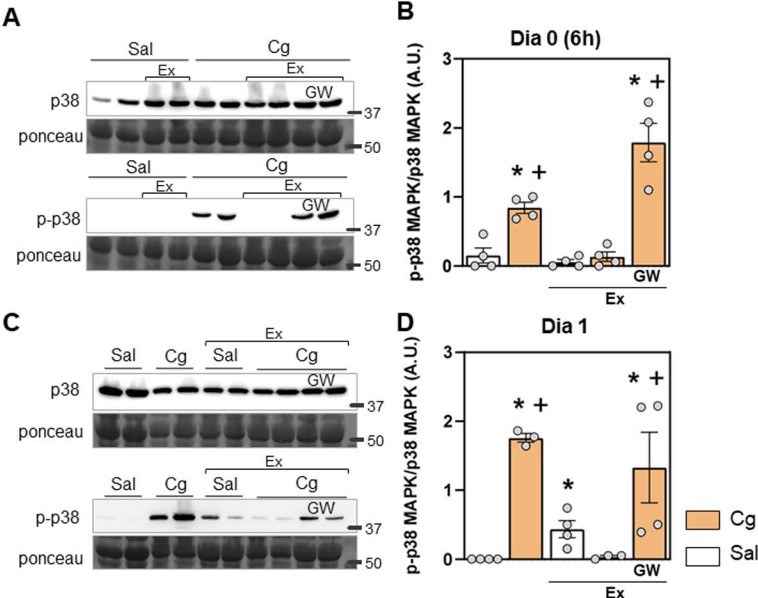

**Fig 6. Effects of physical exercise on p-p38 MAPK expression inskeletal muscle at day 0 and day 1, involves PPARγ receptors. A and C)** Representative image of western blotting membranes demonstrating expression of p38 MAPK, p-p38 MAPK proteins and total protein (Ponceau S) staining on day 0 (6 hours) (A) and day 1 (C), in saline (Sal), exercised (ex) Sal, carrageenan (Cg), exercised Cg and exercised Cg with the PPARγ receptors antagonist strategy, Gw9662 (GW). The number marking indicates the height of the marker in kilodaltons (kDa). (B and D) Quantification of the expression of p-p38 MAPK/p38 MAPK (arbitrary units [A.U.]) on day 0 (6 hours) (B) and day 1 (D) (two-way ANOVA, Tukey post hoc). "*" indicates significant difference (p < 0.05) with Sal and "+" significant difference (p < 0.05) with Cg group. Numbers are presented as Mean ± Standard Error (SE). N = 4 for each group.

showed that carrageenan stimulation led to an increased proportion of macrophages with an inflammatory (M1-like) phenotype, characterized by elevated expression of CD11c and inflammatory cytokines such as IL-1β. Blocking P2X4 receptors reduced M1 polarization and inflammatory gene expression, underscoring the receptor's role in shaping the macrophage response. It is interesting to point out that treatment of RAW 264.7 cells with IL-4, a known anti-inflammatory insult (Gordon and Martinez 2010), prevented the LPS-induced increase in p2x4 receptor gene along with pro-inflammatory genes (*cd86* and *il-1b*), indicating the P2X4 receptor pathways may induce the polarization of macrophages to pro-inflammatory phenotype.

In our previous study, we demonstrated that macrophages are key cells for the onset of persistent inflammatory muscle hyperalgesia. In a mouse model of transition from acute to persistent inflammatory muscle hyperalgesia, we have previously shown that the increase in M1-like macrophages and IL-1β are acute events (from day 1 and at day 2, respectively) [26]. Now we demonstrate that the phosphorylation of p38 MAPK was found to be a downstream effect of P2X4 activation. This was supported by both in vitro and in vivo data, where 5-BDBD treatment significantly reduced p38 MAPK phosphorylation following carrageenan injection. In the context of the inflammatory cascade, p38 MAPK phosphorylation was identified as an acute event occurring within the first 2 days of inflammation. Our in vivo results corroborate the temporal association between P2X4 activation and the onset of macrophage-mediated inflammatory responses, as evidenced by increased p-p38 MAPK levels in muscle tissue. These findings suggest that targeting P2X4 receptors and their downstream signaling pathways could provide a window of opportunity for therapeutic intervention during the acute phase of inflammation.

Although, activation of P2X4 receptor triggers p38 MAPK phosphorylation (p-p38 MAPK) [37] in the early phases of the inflammatory process [11,38], the mechanisms by which P2X4/p-p38 MAPK induced an increase in IL-1β during the acute phase of inflammatory muscle hyperalgesia need to be investigated. In several inflammatory models, it has been shown that an important event of downstream signaling of P2X4 receptor is the activation of the NLRP3 inflammasome [39–41], which is related to activation of caspase-1 and the cleavage of pro-IL-1β to mature IL-1β [42,43]. Therefore, we suggest NLRP3 may be downstream of P2X4 receptor in the transition from acute to persistent inflammatory muscle hyperalgesia.

Physical exercise emerged as a potent modulator of inflammatory signaling pathways. We have showed that regular physical exercise, previous to the inflammatory insults, triggers anti-inflammatory effects, by preventing persistent muscle hyperalgesia and the increase in IL-1β through activation of the peroxisome proliferator-activated receptor gamma (PPARγ), in addition to decreasing M1-like and inducing M2-like macrophages in the muscle tissue [26]. Consistent with our previous findings, regular exercise prevented the transition from acute to persistent inflammatory hyperalgesia, an effect mediated by PPARγ activation. Notably, exercise attenuated p38 MAPK phosphorylation in the acute inflammatory phase, and this effect was reversed by PPARγ receptor antagonism. These results suggest a novel link between PPARγ activation and the inhibition of inflammation-driven p38 MAPK signaling, offering a mechanistic basis for the anti-inflammatory effects of exercise in skeletal muscle.

It is interesting to point out that, in models of diabetes, liver disease and atherosclerosis, physical exercise decreases the activation of NLRP3 [44–46]. Also, PPARγ receptors suppress NLRP3 in neuronal cell culture [37] and repress NF-kB related to NLRP3 [47]. Considering these evidence and our previous and present study, we suggest that physical exercise, through PPARγ receptors, decreases activation of P2X4 induced by an inflammatory stimulus, prevents NF-kB/NLRP3 pathway and reduces M1-macrophage and IL-1β expression. We previously observed that in these conditions, PPARγ receptors are localized in myocytes but not in macrophages [26], therefore, a crosstalk mechanism between skeletal muscle and macrophages to control the transition from acute to persistent inflammatory muscle hyperalgesia should be better explored.

Our findings suggest a multi-faceted regulatory network involving P2X4 receptors, macrophage polarization, p38 MAPK signaling, and PPARγ activation. These pathways appear to converge in the regulation of inflammatory muscle hyperalgesia, providing a coherent framework for understanding the molecular and cellular basis of pain progression in inflammatory conditions. Based on these insights, we propose that:

1. Physical exercise may exert its anti-inflammatory effects by modulating P2X4 – p38 MAPK receptor activity indirectly or directly through PPARγ-mediated mechanisms. Investigating the crosstalk between these pathways could reveal novel targets for anti-inflammatory therapies.

2. The acute phase of inflammation, characterized by p38 MAPK phosphorylation and macrophage polarization, represents a critical period for intervention. Exploring time-dependent effects of P2X4 antagonists or PPARγ agonists could optimize therapeutic efficacy.

3. Combining pharmacological P2X4-related pathways inhibition with exercise-based interventions may yield synergistic effects in preventing the transition to persistent hyperalgesia. Future studies should evaluate this approach in both preclinical and clinical settings.

## 5. Conclusions

In conclusion, our study underscores the central role of P2X4 receptors in macrophage-mediated inflammation and highlights physical exercise mediated PPARγ activation as a non-pharmacological intervention capable of modulating inflammatory signaling pathways. These findings lay the groundwork for future investigations into targeted therapies that harness the complementary effects of pharmacological and lifestyle-based approaches in managing inflammatory muscle pain.

## Supporting information

**S1 Fig. Total protein staining used for western blots normalization.** Representative image of total protein loading stained by Ponceau S dye in the membranes containing skeletal muscle samples of Day 0 (A, B, E and F) and Day 1 (C, D, G and H) after Saline (Sal) or Carrageenan (Cg) injections. B, D, F and G represent the membranes probed with anti-p-p38 antibody. A, C, E and G represent membranes that were probed with anti-p38 antibody. A–D include non-exercise animals with and without the 5-BDBD strategy. E–H include exercised animals with and without GW9662 strategy. The number marking indicates the height of the marker in kilodaltons (kDa). Numbers on the lanes indicate the biological replicates in the gel.
(PDF)

**S2 Fig. Effects of physical exercise on p-p38 MAPK expression in skeletal muscle at day 2 after Inflammation induction.** A–D) Representative image of western blotting membranes demonstrating expression of p38 MAPK, p-p38 MAPK proteins and ponceau staining on day 2, in Control and Exercise (Ex) groups injected with saline (Sal) or carrageenan (Cg). Total protein staining for each membrane are presented by the Ponceau S images. The number marking indicates the height of the marker in kilodaltons (kDa). The numbers on the lanes indicates the biological replicates. Original unedited membranes. Full membranes were imaged through chemiluminescent detection for 10–20 seconds. E) Expression of p-p38 MAPK/p38 MAPK (arbitrary units [A.U.]) on day 2 (two-way ANOVA, Tukey post hoc). "*" indicates difference with Control Sal and Exercise Sal groups, and "+" difference with Exercise Cg group. N = 4 for each group. F) Expression of the cytokine IL-1β in picograms/milligrams of total protein (pg/mg) in the muscle tissue at day 2 after the injection in controls saline (Sal) and Carrageenan (Cg), and in a group that received 5-BDBD previous to Cg. Symbol "**" indicates difference with Sal and Cg groups (one-way ANOVA, Tukey post hoc).
(TIF)

**S3 Fig. p38 MAPK phosphorylation is increased 30 minutes after inflammatory stimulus on macrophages.** Representative image of western blotting membrane demonstrating expression of p-p38 MAPK over time after carrageenan stimulation, shows increased p-p38 MAPK expression at 30 min after inflammatory stimulus.
(TIF)

**S4 Fig. Original blots from** Fig 4. Western blotting membranes demonstrating the expression of p38 MAPK, p-p38 MAPK and endogenous beta tubulin proteins, in RAW 264.7 treated with PBS, carrageenan (Cg), 5-BDBD + Cg or IL-4 + Cg. The number marking indicates the height of the marker in kilodaltons (kDa). Numbers on the lanes indicate biological replicates. Original unaltered images. Full membranes were imaged through chemiluminescent detection for 10–20 seconds.
(TIF)

**S5 Fig. Original blots from** Figs 5 and 6. A–D) Western blotting membranes demonstrating the expression of p38 MAPK, p-p38 MAPK proteins at days 0 (6h) and 1, in gastrocnemius muscle of mice treated with Saline (Sal), carrageenan (Cg) or 5-BDBD + Cg. E–H) Western blotting membranes demonstrating the expression of p38 MAPK, p-p38 MAPK proteins at days 0 (6h) and 1, in gastrocnemius muscle of sedentary and exercised (Ex) mice injected with Sal, Cg or GW9662 (GW) + Cg. The number marking on the side indicates the height of the marker in kilodaltons (kDa). Numbers on the lanes indicate biological replicates. Original unaltered images. Full membranes were imaged through chemiluminescent detection for 10–20 seconds.
(TIF)

## Author contributions

**Conceptualization:** Graciana de Azambuja, Maria Claudia Gonçalves de Oliveira.

**Data curation:** Graciana de Azambuja, Fernando Moreira Simabuco, Maria Claudia Gonçalves de Oliveira.

**Formal analysis:** Graciana de Azambuja, Fernando Moreira Simabuco, Maria Claudia Gonçalves de Oliveira.

**Funding acquisition:** Graciana de Azambuja, Maria Claudia Gonçalves de Oliveira.

**Investigation:** Graciana de Azambuja.

**Methodology:** Graciana de Azambuja, Fernando Moreira Simabuco.

**Project administration:** Graciana de Azambuja, Maria Claudia Gonçalves de Oliveira.

**Resources:** Graciana de Azambuja, Maria Claudia Gonçalves de Oliveira.

**Supervision:** Maria Claudia Gonçalves de Oliveira.

**Validation:** Graciana de Azambuja.

**Visualization:** Graciana de Azambuja, Maria Claudia Gonçalves de Oliveira.

**Writing – original draft:** Graciana de Azambuja.

**Writing – review & editing:** Graciana de Azambuja, Fernando Moreira Simabuco, Maria Claudia Gonçalves de Oliveira.

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
