## [Decision Letter · Decision Letter 0]

15 Nov 2024

PONE-D-24-35584Macrophage-P2X4 receptors pathway is essential to persistent inflammatory muscle hyperalgesia onset, and is prevented by physical exercisePLOS ONE

Dear Dr. Oliveira,

Thank you for submitting your manuscript to PLOS ONE. After careful consideration, we feel that it has merit but does not fully meet PLOS ONE’s publication criteria as it currently stands. Therefore, we invite you to submit a revised version of the manuscript that addresses the points raised during the review process.

We look forward to receiving your revised manuscript.

Kind regards,

Santosh Kumar Mishra

Academic Editor

PLOS ONE

**Journal Requirements:**

São Paulo Research Foundation (FAPESP) – process number: 2018/13599-1; 2020/10585-0; 2021/02921-2

Coordination of Superior Level Staff Improvement (CAPES) – 001

4. Please include your tables as part of your main manuscript and remove the individual files. Please note that supplementary tables (should remain/ be uploaded) as separate "supporting information" files.

Reviewers' comments:

Reviewer's Responses to Questions

**Comments to the Author**

1. Is the manuscript technically sound, and do the data support the conclusions?

Reviewer #1: Yes

Reviewer #2: Yes

2. Has the statistical analysis been performed appropriately and rigorously? 

Reviewer #1: Yes

Reviewer #2: Yes

3. Have the authors made all data underlying the findings in their manuscript fully available?

Reviewer #1: Yes

Reviewer #2: Yes

4. Is the manuscript presented in an intelligible fashion and written in standard English?

Reviewer #1: Yes

Reviewer #2: Yes

5. Review Comments to the Author

**Reviewer #1:**  "The macrophage-P2X4 receptor pathway plays a crucial role in the onset of persistent inflammatory muscle hyperalgesia, which can be mitigated through physical exercise" - upon reviewing this manuscript, I offer the following observations:

1.The paper is composed with clarity and precision.

2.The proposed hypothesis demonstrates considerable strength.

Kindly address the following points in the revised version:

1. - Elucidate the process for preparing the carrageenan solution and specify its concentration.

2. - Provide justification for the exclusive use of male mice in this research.

3. - Describe the nature of the mechanical stimulus applied at the injection site and its duration.

4. - Consider restructuring and enhancing the discussion section for improved clarity and impact.

**Reviewer #2:**  Minor comments:

1. Please provide a reference for the “previously standardized time” mentioned in the following statement: “For protein analysis of p38 MAPK…stimulated with carrageenan (100 mg/mL) for 30 minutes (previously standardized time)”

2. Please provide catalog numbers for antibodies.

3. List out Primer sets for all genes tested with SYBR green. These can be in a supplementary file/table.

4. Which outlier calculation tool was used? GraphPad has several options.

5. Does exogenous ATP (P2X4’s ligand) on Day 6 or 17 cause a significant change in behavior on top of the reported hyperalgesia result?

6. It is difficult to identify what the arrows in figure 3 are pointing at.

Major comments:

1. There are no figure legends available.

2. The authors demonstrate that P2X4 causes an increase in p-p38 MAPK and that exercised mice have a decrease in p-P38 MAPK due to increased PPARγ activation. Is phosphorylation of p38 MAPK affected by 5-BDBD administration in the exercised mice, since P2X4 is proposed to also alter p-p38 MAPK levels? What about if ATP is administered in these mice? The link between the PPARγ and the rest of the paper about P2X4 is not clear. Are these independent pathways that affect p-p39 MAPK levels? Are they parts of the same cell signaling chain that produces musculoskeletal hyperalgesia? Does activation of P2X4 affect PPARγ expression or function?

6. PLOS authors have the option to publish the peer review history of their article (what does this mean? ). If published, this will include your full peer review and any attached files.

**Do you want your identity to be public for this peer review?** For information about this choice, including consent withdrawal, please see our Privacy Policy .

Reviewer #1: **Yes: ** Debjeet Sur

Reviewer #2: No

---

## [Author Response · Author response to Decision Letter 1]

30 Dec 2024

# Response to academic editor’s comments:

1. When submitting your revision, we need you to address these additional requirements. Please ensure that your manuscript meets PLOS ONE's style requirements, including those for file naming.

Manuscript was adjusted to meet PLOS ONE’s style requirements.

São Paulo Research Foundation (FAPESP) – process number: 2018/13599-1; 2020/10585-0; 2021/02921-2; Coordination of Superior Level Staff Improvement (CAPES) – 001.

We state that "The funders had no role in study design, data collection and analysis, decision to publish, or preparation of the manuscript."

3. PLOS requires an ORCID iD for the corresponding author in Editorial Manager on papers submitted after December 6th, 2016. Please ensure that you have an ORCID iD and that it is validated in Editorial Manager.

When trying to link the ORCID iD for the corresponding author we received a message stating that the ORCID iD already existed. Please, let us know whether this is a bug that we can fix it.

4. Please include your tables as part of your main manuscript and remove the individual files. Please note that supplementary tables (should remain/ be uploaded) as separate "supporting information" files.

Tables were included in the main manuscript.

5. Please include captions for your Supporting Information files at the end of your manuscript, and update any in-text citations to match accordingly.

Captions for main and supporting information files were included at the end of the manuscript. In-text citations were revised and corrected.

6. PLOS ONE now requires that authors provide the original uncropped and unadjusted images underlying all blot or gel results reported in a submission’s figures or Supporting Information files.

Uncropped and unadjusted original blot results for all assays are included as supporting information files with captions.

7. Please review your reference list to ensure that it is complete and correct.

Reference list was revised and no changes were made. Citation style was adapted to Plos One format.

# Response to reviewer’s comments:

Reviewer #1: "The macrophage-P2X4 receptor pathway plays a crucial role in the onset of persistent inflammatory muscle hyperalgesia, which can be mitigated through physical exercise" - upon reviewing this manuscript, I offer the following observations:

1.The paper is composed with clarity and precision; 2.The proposed hypothesis demonstrates considerable strength.

Thank you for your positive feedback regarding the clarity and precision of our manuscript and the strength of the proposed hypothesis. We deeply appreciate your encouraging remarks, which affirm the efforts we put into developing and presenting our work.

Kindly address the following points in the revised version:

1. Elucidate the process for preparing the carrageenan solution and specify its concentration.

The revised information has been updated and included in the revised manuscript under [Procedures for intramuscular injections, drugs, and doses, page 6]: Lambda Carrageenan was freshly prepared at a concentration of 1 mg/mL and subsequently diluted to a working solution of 5 µg/µL in 0.9% isotonic saline.

2. Provide justification for the exclusive use of male mice in this research.

Thank you for raising this important point. The exclusive use of male mice in our research was based on the fact that both the exercise model and the hyperalgesia model were initially validated in male mice as part of our initial exploration. However, we recognize the importance of understanding sex-based differences. Our group is in the process of submitting a new paper that includes the validation of these approaches in female mice and explores differences in molecular mechanisms involved in exercise and muscle hyperalgesia.

3. Describe the nature of the mechanical stimulus applied at the injection site and its duration.

The nature of the mechanical stimulus is pressure. Mechanical stimulation was conducted using an analgesimeter, which applies gradually increasing pressure to the injection site. The device's concave tip delivers mechanical stimuli that target deep tissues, including skeletal muscle and fascia. This is clarified in the revised manuscript under [Mechanical testing of muscle nociceptive threshold to assess the muscle hyperalgesia, page 6].

4. Consider restructuring and enhancing the discussion section for improved clarity and impact.

We have incorporated the comments from Reviewer #1 and Reviewer #2 and have made improvements to the manuscript's structure, particularly in the Discussion section. We believe these revisions enhance the clarity of the results interpretation, strengthen the presentation of new hypotheses and open questions, and ultimately improve the impact of the study.

Reviewer #2:

Minor comments:

1. Please provide a reference for the “previously standardized time” mentioned in the following statement: “For protein analysis of p38 MAPK…stimulated with carrageenan (100 mg/mL) for 30 minutes (previously standardized time)”

An initial experiment was conducted in cells to determine the phosphorylation timing of the p38 MAPK protein upon stimulation with Carrageenan at a concentration of 100 mg/mL. This information has been included in the revised manuscript under and an additional figure has been provided as a supplementary information file ([Sup Fig. 3]).

2. Please provide catalog numbers for antibodies.

This information is updated in the revised manuscript under [Procedure with primary culture of peritoneal macrophages, page 8-9; and Procedures with RAW 264.7 macrophage culture, page 11].

3. List out Primer sets for all genes tested with SYBR green. These can be in a supplementary file/table.

List of primer sets were added in a table in the manuscript under [Procedures with RAW 264.7 macrophage culture, page 10].

4. Which outlier calculation tool was used? GraphPad has several options.

Indeed, outliers were calculated using GraphPad Quick calc tool which performs the Grubbs' test. Significance level was set to 0.05. This information is updated in the manuscript under [Statistical analysis, page 13].

5. Does exogenous ATP (P2X4’s ligand) on Day 6 or 17 cause a significant change in behavior on top of the reported hyperalgesia result?

These experiments have not been performed in the current study. While outside the scope of this research, our group has previously tested the administration of a non-hydrolyzable form of ATP (α,β-meATP) in muscle and observed that it induces muscle hyperalgesia by activating P2X3 receptors, while P2X4 receptors were not evaluated in this study (Schiavuzzo et al., Neuroscience, 2015 Jan 29:285:24-33). Furthermore, prior studies have demonstrated that exogenous ATP can trigger muscle hyperalgesia and is involved in the molecular processes that lead to muscle hyperalgesia in rodent models. Previous research has shown that α,β-meATP or a metabolite cocktail (combining protons, lactate, and ATP) can induce skeletal muscle hyperalgesia in rodent models (Gregory et al., 2015, PLoS One. 2015 Sep 17;10(9):e0138576; Pollak et al., 2014, Exp Physiol. 99(2):368-80). Additionally, it we provide evidence that carrageenan-induced muscle hyperalgesia is mediated in part by P2X4 receptors, suggesting that there is an increase on ATP concentration. Based on this background, we anticipate that in the present model, administration of exogenous ATP would likely exacerbate the reported muscle hyperalgesia.

6. It is difficult to identify what the arrows in figure 3 are pointing at.

Figure 3 was updated and we expect to be improved. We included an inset for each treatment condition, to highlight the expression of the cell markers. Updated version is found as figure 3.

Major comments:

1. There are no figure legends available.

We acknowledge that the figure legends were not positioned correctly in the original manuscript. In the revised version, the figure legends have been relocated to the end of the manuscript, following the reference list, in accordance with PLOS ONE’s manuscript formatting guidelines.

2. The authors demonstrate that P2X4 causes an increase in p-p38 MAPK and that exercised mice have a decrease in p-P38 MAPK due to increased PPARγ activation. Is phosphorylation of p38 MAPK affected by 5-BDBD administration in the exercised mice, since P2X4 is proposed to also alter p-p38 MAPK levels? What about if ATP is administered in these mice? The link between the PPARγ and the rest of the paper about P2X4 is not clear. Are these independent pathways that affect p-p39 MAPK levels? Are they parts of the same cell signaling chain that produces musculoskeletal hyperalgesia? Does activation of P2X4 affect PPARγ expression or function?

Thank you for these insightful questions. We present a point-by-point response for discussion with the reviewer and when appropriate, we have addressed them in the revised manuscript at the discussion section, as follows:

1. Effect of 5-BDBD on p-p38 MAPK in exercised mice: Administration of 5-BDBD in exercised mice was not specifically analyzed for its effect on p-p38 MAPK levels in the current study. Based on the proposed pathway, we hypothesize that exercise pathways are inhibiting P2X4 and would prevent the phosphorylation of p38 MAPK. Injecting 5-BDBD in exercised mice could potentiate exercise effects on the P2X4 pathways. However, we did not proposed to test a synergistic effect of the antagonist and the exercise, thus this subject could become a topic for future experiments.

2. Effect of ATP administration on p-p38 MAPK in exercised mice: The effect of ATP administration on p-p38 MAPK levels in exercised mice was not directly assessed in our study. While we did not explore ATP levels in skeletal muscle tissue, either in sedentary or exercised mice, we hypothesize that ATP administration could enhance p-p38 MAPK phosphorylation, increase inflammatory responses, and exacerbate persistent skeletal muscle hyperalgesia in this model. Importantly, we speculate that exercised mice may have improved metabolic function and potentially greater resilience to inflammatory processes or the effects of inflammatory products like ATP. However, further studies are needed to confirm this hypothesis.

3. Clarifying the link between PPARγ and P2X4 in the context of p-p38 MAPK: We agree that the connection between PPARγ and P2X4 requires further clarification, and our study does not fully address this relationship. In the revised manuscript discussion section, we emphasize that PPARγ activation by exercise appears to mitigate p38 MAPK phosphorylation, potentially counteracting the effects of P2X4 activation. While both exercise-activated PPARγ and carrageenan-induced P2X4 converge on p-p38 MAPK levels, it remains unclear whether they function independently, repressing each other, or are part of a shared signaling cascade. We propose that skeletal muscle from exercised mice exhibits increased PPARγ activation, which may contribute to greater resilience to inflammatory insults. The underlying mechanisms require further investigation. Specifically, we hypothesize that exercise-induced PPARγ activation could either enhance the anti-inflammatory response in muscle tissue, reducing P2X4-p38 MAPK signaling in inflammatory macrophages, or directly/indirectly repress P2X4 activation in macrophages cells, thus preventing inflammation exacerbation.

4. Impact of P2X4 activation on PPARγ expression or function: To date, there is no direct evidence in our study, or others, that P2X4 activation affects PPARγ expression or function. This remains an open question and we may include this in future research to better understand the interplay between these pathways in musculoskeletal hyperalgesia.

---

## [Editor Report · Decision Letter 1]

10 Jan 2025

Macrophage-P2X4 receptors pathway is essential to persistent inflammatory muscle hyperalgesia onset, and is prevented by physical exercise

PONE-D-24-35584R1

Dear Dr. Gonçalves de Oliveira,

We’re pleased to inform you that your manuscript has been judged scientifically suitable for publication and will be formally accepted for publication once it meets all outstanding technical requirements.

Kind regards,

Santosh Kumar Mishra

Academic Editor

PLOS ONE
---

## [Editor Report · Acceptance letter]

PONE-D-24-35584R1

PLOS ONE

Dear Dr. Gonçalves de Oliveira,

I'm pleased to inform you that your manuscript has been deemed suitable for publication in PLOS ONE. Congratulations! Your manuscript is now being handed over to our production team.

Kind regards,

on behalf of

Dr. Santosh Kumar Mishra

Academic Editor

PLOS ONE